# Improving Neural Machine Translation by Multi-Knowledge Integration with Prompting

**Ke Wang, Jun Xie**[*]**, Yuqi Zhang, Yu Zhao**
Alibaba Group
{wk258730,qingjing.xj,chenwei.zyq}@alibaba-inc.com,kongyu@taobao.com

## Abstract

Improving neural machine translation (NMT) systems with prompting has achieved significant progress in recent years. In this work, we focus on how to integrate multi-knowledge, multiple types of knowledge, into NMT models to enhance the performance with prompting. We propose a unified framework, which can integrate effectively multiple types of knowledge including sentences, terminologies/phrases and translation templates into NMT models. We utilize multiple types of knowledge as prefix-prompts of input for the encoder and decoder of NMT models to guide the translation process. The approach requires no changes to the model architecture and effectively adapts to domain-specific translation without retraining. The experiments on English-Chinese and English-German translation demonstrate that our approach significantly outperform strong baselines, achieving high translation quality and terminology match accuracy.

## 1 Introduction

In the workflow of translation, human translators generally utilize different types of external knowledge to simplify the process and improve translation quality and speed, such as matching terminologies and similar example sentences. The knowledge used in machine translation mainly includes high-quality bilingual sentences, a bilingual terminology dictionary and translation templates. Intuitively, it is reasonable to believe that it is beneficial for improving translation quality to integrate multiple types of knowledge into NMT models in a flexible and efficient way. However, most existing methods focus on only how to integrate a single type of knowledge into NMT models, either a terminology dictionary (Dinu et al., 2019; Dougal and Lonsdale, 2020), bilingual sentences (Cao and

---

[*]Corresponding author.

Xiong, 2018; Liu et al., 2019a) or translation templates (Yang et al., 2020).

As a primary technique to utilize a terminology dictionary, lexically constrained translation allows for explicit phrase-based constraints to be placed on target output strings (Hu et al., 2019). Several research works (Hokamp and Liu, 2017; Post and Vilar, 2018) impose lexical constraints by modifying the beam search decoding algorithm. Another line of approach trains the model to copy the target constraints by data augmentation (Song et al., 2019; Dinu et al., 2019; Chen et al., 2020). Some researchers (Li et al., 2019; Wang et al., 2022b) introduce attention modules in the architecture of NMT models to integrate constraints. These methods using terminologies or phrases as the knowledge suffer from either high computational overheads or low terminology translation success rates.

In the majority of methods that utilize sentence pairs, the most similar source-target sentence pairs are retrieved from a translation memory (TM) for the input source sentence (Liu et al., 2019a; Huang et al., 2021; He et al., 2021). Several approaches focus on integrating a TM into statistical machine translation (SMT) (Ma et al., 2011; Wang et al., 2013; Liu et al., 2019b). Some researchers use a TM to augment an NMT model, including using n-grams from a TM to reward translation (Zhang et al., 2018b), employing an auxiliary network to integrate similar sentences into the NMT (Gu et al., 2018; Xia et al., 2019) and data augmentation based on TM (Bulte and Tezcan, 2019a; Xu et al., 2020). These methods consume considerable computational overheads in training or testing.

Although these approaches have demonstrated the benefits of combining an NMT model with a single type of knowledge, how to integrate multiple types of knowledge into NMT models remains a challenge. In this work, we propose a prompt-based neural machine translation that can integrate multiple types of knowledge including both sen-

tences, terminologies/phrases and translation templates into NMT models in a unified framework. Inspired by (Brown et al., 2020), which has redefined different NLP tasks as fill in the blanks problems by different prompts, we concatenate the source and target side of the knowledge as prefix-prompts of input for the encoder and decoder of NMT models, respectively. During training, this model learns dynamically to incorporate helpful information from the prefixes into generating translations. At inference time, new knowledge from multiple sources can be applied in real time. The model has automatic domain adaption capability and can be extended to new domains without updating parameters. We evaluate the approach in two tasks domain adaptation and soft lexical (terminology) constraint. The metric of 'exact match' for terminology match accuracy has significantly improved compared to strong baselines both in English to German and English to Chinese translation. This approach has shown its robustness in domain adaptation and performs better than fine-tuning when there are domain mismatch or noise data.

The contributions of this work include:

- We propose a simple and effective approach to integrate multi-knowledge into NMT models with prompting.

- We demonstrate that an NMT model can benefit from multiple types of knowledge simultaneously, including sentence, terminology/phrases and translation template knowledge.

## 2 Related Work

NMT is increasingly improving translation quality. However, the interpolation of the reasoning process has been less clear due to the deep neural architectures with hundreds of millions of parameters. How to guide an NMT system with user-specified different types of knowledge is an important issue of NMT applications in real world.

The first and most studied knowledge is simple constraints such as lexical constraints or in-domain dictionaries. (Hokamp and Liu, 2017) proposes grid beam search (GBS) by modifying the decoding algorithm to add lexical constraints. (Post and Vilar, 2018) introduces dynamic beam allocation to reduce the runtime complexity of GBS by dividing a fixed size of beam for candidates. (Hu

et al., 2019) proposes vectorized dynamic beam allocation (VDBA) to improve the efficiency of the decoding algorithm further. The beam search decoding algorithm by adding lexical constraints is still significantly slower than the beam search algorithm. Some data augmentation works propose to replace the corresponding source phrases with the target constraints (Song et al., 2019), to integrate constraints as inline annotations in the source sentence (Dinu et al., 2019), to insert target constraints using an alignment model (Chen et al., 2021) and to append constraints after the source sentence with a separation symbol (Chen et al., 2020; Jon et al., 2021). These data augmentation methods can not guarantee the presence of the target constraints in the output.

Some works concentrate on adapting the architecture of NMT models to add lexical constraints. (Susanto et al., 2020) invokes lexical constraints using a non-autoregressive decoding approach. (Zhang et al., 2021) introduces explicit phrase alignment into the translation process of NMT models by building a search space similar to phrase-based SMT. (Li et al., 2019) proposes to use external continuous memory to store constraints and integrate the constraint memories into NMT models through the decoder network. (Wang et al., 2022a) proposes a template-based method for constrained translation while maintaining the inference speed. (Wang et al., 2022b) proposes to integrate vectorized lexical source and target constraints into attention modules of the NMT model to model constraint pairs. These methods may still suffer from low match accuracy of terminology when decoding without the VDBA algorithm.

The use of TM is very necessary for computer-aided translation (Yamada, 2011) and computational approaches for machine translation (Koehn and Senellart, 2010). Similar sentence pairs retrieved from a TM are also utilized as a type of knowledge to enhance the translation (Liu et al., 2019a; He et al., 2021; Khandelwal et al., 2021). (Farajian et al., 2017) exploits the retrieved sentence pairs from a TM to update the generic NMT models on-the-fly. (Zhang et al., 2018b) utilizes translation pieces based on n-grams extracted from a TM during beam search by adding rewards for matched translation pieces into the NMT model output layer. (He et al., 2019) proposes to add the word position information from a TM as additional rewards to guide the decoding of NMT models.

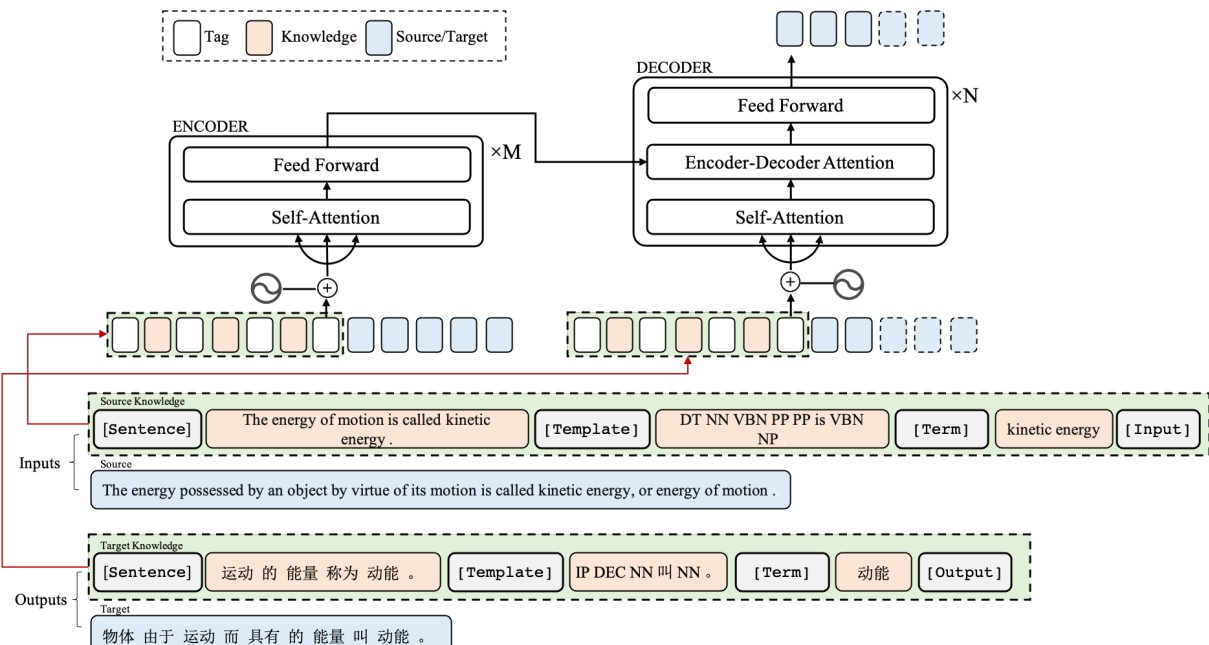

Figure 1: Knowledge integration framework with prompting and an example representation.

(Gu et al., 2018) uses an auxiliary network to fuse information from the source sentence and the retrieved value from a TM and then integrate it into the NMT model architecture. (Xia et al., 2019) proposes to pack a TM into a compact graph corresponding to multiple words for different sentences in a TM, and then encode the packed graph into a deep representation during the decoding phase. (Xu et al., 2020) utilizes data augmentation to train an NMT model whose training instances are bilingual sentences augmented with the translation retrieved from the TM. For input sentences that are not very similar to their TMs, the translation performance of these methods suffers significantly. (He et al., 2021) introduces Example Layer consisting of multi-head attention and cross attention to translate any input sentences whether they are similar to their TM or no. (Cai et al., 2021) extends the TM from the bilingual setting to the monolingual setting through learnable memory retrieval in a cross-lingual manner. The key idea of TM is to integrate the retrieved sentence pairs from a TM into the NMT architecture for accurate translations. Most of the works integrate the TM knowledge via model modification and the models need to be retrained when loading another TM in new domains.

The general knowledge integration to NMT is an ongoing work (Tang et al., 2016; Liu et al., 2016; Zhang et al., 2018a). (Yang et al., 2020) proposes to use extracted templates from tree struc-

tures as soft target templates to incorporate the template information into the encoder-decoder framework. (Shang et al., 2021) introduce to a template-based machine translation (TBMT) model to integrate the syntactic knowledge of the retrieved target template in the NMT decoder. (Zhang et al., 2018a) represents prior knowledge sources as features in a log-linear model to guide the learning process of NMT models. These approaches have demonstrated the clear benefits by incorporating different single types of knowledge into NMT models. Our approach is designed to integrate multiple types of knowledge into NMT models through an unified framework.

## 3 Approach

As shown in Figure 1, we use source-side knowledge sequence and target-side knowledge sequence to prepend a source sentence and a target sentence as a prefix separately. The model uses multiple types of knowledge as prefix-prompts of the input to guide the process of translating target sentence. The form is intended to lead the NMT model how to utilize relevant information from the redundant prefixes to guide the translation process for improving translation quality. We use three types of special tokens to separate different types of knowledge sequence, source sentence and target sentence.

- [Sentence]/[Term]/[Template]: It indicates similar sentences, matching termi-

nologies and translation templates respectively. The first token of each knowledge sequence is always the special token.

- `[Input]`: It is used to separate the knowledge sequence and the source sentence.

- `[Output]`: It is used to separate the knowledge sequence and the target sentence.

For each sentence pair $\langle \mathbf{x}, \mathbf{y} \rangle$, corresponding similar sentence pairs, matching terminologies and translation templates are concatenated into source knowledge sequence $\mathbf{x_k}$ and target knowledge sequence $\mathbf{y_k}$ with the corresponding special token `[Sentence]`, `[Term]` and `[Template]` on the source and target side, respectively. Then the sentence pair $\langle \mathbf{x}, \mathbf{y} \rangle$ are preprocessed as follows: the source sentence $\mathbf{x}$ and target sentence $\mathbf{y}$ are concatenated with source knowledge sequence $\mathbf{x_k}$ and target knowledge sequence $\mathbf{y_k}$, respectively. An Example of the format of the input and output sequences is given in Figure 1. When similar sentences and matching terminologies retrieved are empty, the input sequence and output sequence contain only translation template knowledge sequence. Although in this paper we integrate similar sentences, terminologies and translation templates into the NMT models, our approach can utilize more types of knowledge to improve translation performance by using this labeling strategy.

### 3.1 Training

Given a source sentence $\mathbf{x}$, the conditional probability of the corresponding target sentence $\mathbf{y}$ by incorporating source and target knowledge sequence $\langle \mathbf{x_k}, \mathbf{y_k} \rangle$ is defined as follows:

$$P(\mathbf{y}|\mathbf{x}, \mathbf{x_k}, \mathbf{y_k}; \theta) = \prod_{i=1}^{n} P(\mathbf{y_j}|\mathbf{x}, \mathbf{y_{<j}}, \mathbf{x_k}, \mathbf{y_k}; \theta),$$
(1)

where $\theta$ is a set of model parameters, $\mathbf{y_{<j}} = y_1, ..., y_{j-1}$ denotes a sequence of translation prefix tokens at time step $j$ and $n$ is length of the target sentence $\mathbf{y}$.

Similar to the vanilla NMT, we use the maximum likelihood estimation (MLE) loss function to find a set of model parameters on training set $\mathcal{D}$. In order to focus the model on learning the target sentence, we utilize only tokens from the target sentence to calculate the loss function instead of the whole output sentence that contains knowledge sequence. Formally, we minimize the following loss function:

$$\mathcal{L} = \frac{1}{|\mathcal{D}|} \sum_{(\mathbf{x},\mathbf{y}) \in \mathcal{D}} - \log P(\mathbf{y}|\mathbf{x}, \mathbf{x_k}, \mathbf{y_k}; \theta). \quad (2)$$

Note that the proposed method is different from the priming method (Bulte and Tezcan, 2019b; Pham et al., 2020). The priming techniques retrieve similar source sentences and corresponding translations, and then the similar sentence pairs can be used as an input prompt for NMT models. First, we train the model based on our proposed loss function Equation 2 and the NMT model is trained with standard loss function in their works. Also, our method can be applied to multi-knowledge and their work can only be is limited to sentences.

For model optimization, we adopt a two-stage training strategy. In the first stage, we train the standard NMT model based on the standard training objective using the original training data set. Then, in the second stage, we use a training data set constructed from multiple types of knowledge to learn the model parameters based on Equation 2. The proposed model can also be initialized with pre-trained NMT models and then trained on the training data set that contains multi-knowledge.

### 3.2 Inference

The model receives the whole input sequence and the target prefix composed of target knowledge sequence during decoding. Before beginning translation sequence generation, the encoder encodes the input sequence, and the decoder encodes the target prefix. The initial steps of the beam search use the given prefix $\mathbf{y_k}$ to decode the tokens after the special separator token `[Output]` in forced decoding mode. The decoder can gain indirect access to whole input sequence tokens while also gaining direct access to target prefix tokens by self-attention and cross-attention mechanisms. It enables the NMT model learn to how to extract and make use of valuable information from the redundant prefixes during training, and use the prefixes to guide the translation process during inference.

### 3.3 Knowledge Acquisition

We describe the methods employed in this work to how to obtain knowledge from bilingual sentences (sentence knowledge), terminology dictionaries (terminology knowledge) and translation templates (template knowledge).

**Retrieving Similar Sentence** For each source sentence $\mathbf{x}$, we retrieve the most similar bilingual sentences $\langle \mathbf{x_s}, \mathbf{y_s} \rangle$ from sentence knowledge. We use token-based edit distance (Levenshtein, 1965) to calculate the similarity score. Formally, for a given source sentence $\mathbf{x}$, similarity score $\text{sim}(\mathbf{x}, \mathbf{x_s})$ between two source sentences $\mathbf{x}$ and $\mathbf{x_s}$:

$$\text{sim}(\mathbf{x}, \mathbf{x_s}) = 1 - \frac{ED(\mathbf{x}, \mathbf{x_s})}{max(|\mathbf{x}|, |\mathbf{x_s}|)}, \qquad (3)$$

where $ED(\mathbf{x}, \mathbf{x_s})$ denotes the Edit Distance between $\mathbf{x}$ and $\mathbf{x_s}$, and $|\mathbf{x}|$ is the length of $\mathbf{x}$. $\mathbf{x_s}$ is a source sentence from the sentence knowledge. Each source sentence $\mathbf{x}$ is compared to all the sources from the sentence knowledge using the similarity score. We ignore perfect matches and keep the single best match sentence pair if its similarity score is higher than a specified threshold $\lambda$.

**Matching Terminology** Terminology dictionaries specify phrase-level corresponding relationships of a sentence pair. When two matching terminologies from a sentence pair have overlapping ranges, a 'hard' match selects only one of them as matching results, such as maximum matching using the longest matching terminologies. The match strategy causes boundary errors, which could negatively impact the quality of the translation. Therefore, we adopt a 'soft' match strategy to utilize all matching terminologies from sentence pairs. For each source $\mathbf{x}$ and the corresponding target $\mathbf{y}$, we record any matching bilingual terminologies that are fully contained in both the source $\mathbf{x}$ and target $\mathbf{y}$. For each sentence pair, source tokens of the matching terminologies are concatenated into $\mathbf{x_{tm}}$ with the special token `[Term]`, and corresponding target tokens are similarly concatenated into $\mathbf{y_{tm}}$. The NMT model learns to automatically choose the proper terminologies based on the terminological context by redundant prefixes that contain the all matching bilingual terminologies.

**Translation Template Prediction** To construct translation template sequence $\mathbf{x_{tp}}$, we follow (Yang et al., 2020) to extract templates from a sub-tree by pruning the nodes deeper than a specific depth on the sentence corresponding constituency-based parser tree. We gain a parallel training data using the source sentences, extracted source and target templates. The constructed data is employed to train a sequence generation model to predict target template sequence. The model is to take the source sentence and corresponding source template sequence as inputs and generate template sequence as outputs.

## 4 Experiments

In this section, we validate the effectiveness of the proposed approach on translation quality and terminology match accuracy by comparing with the previous methods used only a single type of knowledge. We evaluate translation quality with the case-insensitive detokenized SacreBLEU score (Post, 2018) and terminology match accuracy with exact match accuracy (Anastasopoulos et al., 2021) which is defined as the ratio between the number of matched source term translations in the output and the total number of source terms.

### 4.1 Setup

**Corpus** We evaluate our approach on English-Chinese (En-Zh) and English-German (En-De) translation tasks. For English-German, we use the WMT16 dataset as the training corpus of our model, consisting of 4.5M sentence pairs. We randomly divided the corpus into 4,000 sentences for the validation set and the rest for training. For English-Chinese, we train our model on CCMT2022 Corpus, containing 8.2M sentence pairs. The WMT newsdev2017 is used as the validation set.

We measure the effectiveness of our model on multi-domain test sets. For English-German, we use the multi-domain English-German parallel data (Aharoni and Goldberg, 2020) as in-domain test sets, which include IT, Medical, Koran, and Law. For English-Chinese, We use the multi-domain English-Chinese parallel dataset (Tian et al., 2014) as in-domain test sets, including Subtitles, News and Education. To distinguish the multi-domain sets for testing and the WMT16 or CCMT2022 sets for training, we call the multi-domain datasets as the in-domain training set or in-domain test set. We use only the training sets to train the models and evaluate results on in-domain test sets in our experiments. We retrieve similar sentence pairs from corresponding in-domain training sets for each in-domain test set. We used randomly selected 2,000 sentences from UM-Corpus as the validation sets of Fine-Tuning models. The sentence statistics of datasets are illustrated in Table 1.

For all datasets, we tokenize English and German text with Moses [1] and the Chinese text with

---

[1] https://github.com/moses-smt/mosesdecoder

| Task | Domain | Train | Vaild | Test |
|------|--------|-------|-------|------|
| En-Zh | Subtitles | 298K | 2,000 | 579 |
| | News | 448K | 2,000 | 1,500 |
| | Education | 448K | 2,000 | 790 |
| En-De | IT | 223K | 2,000 | 2,000 |
| | Medical | 248K | 2,000 | 2,000 |
| | Law | 467K | 2,000 | 2,000 |
| | Koran | 18K | 2,000 | 2,000 |

Table 1: The number of training, validation, and test data sets of English-German and English-Chinese multi-domains.

Jieba [2] tokenizer. We train a joint Byte Pair Encoding (BPE) (Sennrich et al., 2016) with 32k merge operations and use a joint vocabulary for both source and target text. The models in all experiments follow the state-of-the-art Transformer base architecture (Vaswani et al., 2017) implemented in the Fairseq toolkit (Ott et al., 2019). The models are trained on 4 NVIDIA V100 GPUs and optimized with Adam algorithm (Kingma and Ba, 2015) with $\beta_1 = 0.9$ and $\beta_2 = 0.98$. We set the learning rate to 0.0007. In all experiments, the dropout rate is set to 0.3 for English-German and 0.1 for English-Chinese. We use early stopping with a patience of 30 for all experiments. We averaged the last 5 checkpoints in all testing.

**Baseline** We compare our approach with the following representative baselines:

- **Vanilla NMT** (Vaswani et al., 2017): We directly train a standard Transformer base model using the training set.

- **Fine-Tuning**: The model is fine-tuned using each in-domain training data set based on vanilla NMT. As a single-domain model with an upper bound on the performance, it loses the ability of multi-domain adaption.

- $k$**NN-MT** (Khandelwal et al., 2021): The non-parametric method combines a NMT model with token-level $k$-nearest-neighbor($k$NN) by retrieving relevant token examples. It uses an in-domain training data set for domain adaptation tasks without additional training. The datastore is generated by an in-domain training set.

- **Priming-NMT** (Pham et al., 2020): It only uses similar sentences as prefixes of a NMT

|  | En-De | En-Zh |
|--|-------|-------|
| Sentence | 33.57% | 39.28% |
| Terminology | 42.66% | 18.23% |

Table 2: Percentage of source sentences with similar sentences and with matching terminologies on the training sets.

model to force the model to generate a translation.

- **VecConstNMT** (Wang et al., 2022b): It vectorizes and integrates lexical constraints (matching terminologies) into NMT models by attention modules. The method outperforms several strong baselines, including the works (Song et al., 2019; Chen et al., 2021).

**Training Data** The similar sentence pairs are extracted from an training data set using a specified similarity threshold of 0.4 in our experiments. For terminology knowledge, we extract a bilingual terminology dictionary from the training data set using a term extraction tool TM2TB [3] with default parameters and use the dictionary to match each source and target sentences in the training data set by the 'soft' match strategy. We use Stanford parser (Manning et al., 2014) to generate source and target templates based on a specific depth 4 from the training data. We build our method's training data by combining corresponding the similar sentence pair, matching terminologies and translation templates for each sentence pair from the training data set. Table 2 provides the percentage of source sentences with a similar sentence pair where the score is higher than the similarity threshold of 0.4 and the percentage of sentences with matching terminologies in the training data set. We use the vanilla NMT to train our proposed models based on the training data.

**Test Data** For each test set of multi-domain sets, we retrieve similar sentence pairs from the training data set and corresponding in-domain training data set. The number of terminologies extracted by TM2TB with the default threshold of 0.9 is not sufficient to validate terminology accuracy in the in-domain test sets. Therefore, we use TM2TB with a similarity threshold of 0.7 to extract the bilingual terminologies from the in-domain test set, and then

[2]https://github.com/fxsjy/jieba

[3]https://github.com/luismond/tm2tb

| Metric | Method (*Knowledge*) | English-German | | | | | English-Chinese | | | |
| | | IT | Medical | Law | Koran | Avg. | Subtitles | News | Education | Avg. |
|---|---|---|---|---|---|---|---|---|---|---|
| BLEU | Fine-Tuning | 40.79 | 53.14 | 56.68 | 28.08 | 44.67 | 27.53 | 33.91 | 47.96 | 36.47 |
| | Vanilla NMT | 23.07 | 30.72 | 35.57 | 10.16 | 24.88 | 24.34 | 31.43 | 38.54 | 31.44 |
| | VecConstNMT (*Term.*) | 23.74 | 30.41 | 35.19 | 10.21 | 24.89 | 24.70 | 31.61 | 38.78 | 31.70 |
| | Priming-NMT (*Sent.*) | 25.93 | 37.94 | 41.28 | 11.41 | 29.14 | 38.79 | 34.83 | 52.83 | 42.15 |
| | *k*NN-MT (*Sent.*) | 31.00 | 45.59 | 50.70 | 17.66 | 36.23 | 41.31 | 35.07 | 51.32 | 42.57 |
| | **Ours** (*Term.+Sent.+Temp.*) | 32.43 | 45.14 | 50.00 | 18.89 | 36.62 | 40.13 | 37.87 | 55.25 | 44.42 |
| Exact Match | Fine-Tuning | 59.41 | 69.70 | 66.29 | 35.45 | 57.71 | 53.25 | 58.29 | 65.89 | 59.14 |
| | Vanilla NMT | 34.09 | 41.92 | 43.29 | 19.09 | 34.60 | 59.17 | 55.50 | 60.75 | 58.47 |
| | VecConstNMT (*Term.*) | 80.91 | 83.99 | 83.87 | 77.27 | 81.51 | 92.31 | 88.01 | 94.39 | 91.57 |
| | Priming-NMT (*Sent.*) | 37.48 | 50.00 | 49.38 | 15.45 | 38.08 | 66.86 | 56.32 | 66.82 | 63.33 |
| | *k*NN-MT (*Sent.*) | 43.56 | 61.15 | 61.16 | 24.55 | 47.61 | 50.30 | 47.87 | 63.55 | 53.91 |
| | **Ours** (*Term.+Sent.+Temp.*) | 80.20 | 86.58 | 89.52 | 81.82 | 84.53 | 92.90 | 94.66 | 90.65 | 92.74 |

Table 3: Evaluation results on the English-German and English-Chinese multi-domain test sets, reported on BLEU and exact match accuracy of terminology. *Term.*, *Sent.* and *Temp.* indicate terminology, sentence and template knowledge, respectively.

use the terminologies to match each sentence pair. We train a model based on the pre-trained model mBART (Liu et al., 2020) using source sentences, source and target templates extracted from parse tree on in-domain training data. Then we use the model to predict target templates of in-domain test data.

## 4.2 Main Results

Table 3 shows the BLEU and exact match accuracy on Fine-Tuning, NMT, VecConstNMT, *k*NN-MT, Priming-NMT, and our proposed method on the English-German and English-Chinese multi-domain test data sets. Our method outperforms all baselines in terms of exact match accuracy on average, demonstrating the benefits of integrating sentence, terminology and template knowledge into NMT models. On English-German multi-domain test sets, our method improves an average of 10.74 BLEU and 49.93% exact match accuracy over vanilla NNT. Compared with Priming-NMT using sentence knowledge, our method enhances performance by up to 7.48 BLEU on average. Our method outperforms than *k*NN-MT in the IT and Koran domains.

For English-Chinese multi-domain test sets, our method performs better than Fine-Tuning. The Subtitles, News and Education three domains training data contains noises, such as sentence pairs or domain mismatches. The performance improvement of Fine-Tuning relied on the quality of the in-domain the training data is not significant. Similarly, the performance of *k*NN-MT depends on training data quality, and our method achieves bet-

ter BLEU in News and Education domains compared to *k*NN-MT. Therefore, our approach has stronger generalization ability and significant performance improvements in these domains compared to the baselines.

## 4.3 Ablation Studies

In this subsection, we perform ablation experiments on proposed models in order to better understand their relative importance. Table 4 shows evaluation results of the proposed model using only one or two type of knowledge on in-domain test sets. Our proposed method (*Sent.*) using sentence knowledge outperforms the strong baseline Priming-NMT by 5.11 BLEU on English-German on average, which indicates that the loss function Equation 2 could significantly enhance the performance during training. Our method (*Term.*) using terminology knowledge outperforms the strong baseline VecConstNMT in terms of exact match accuracy on average.

Our method (*Term.+Sent.*) using sentence and terminology knowledge achieves both BLEU and exact match accuracy improvements compared with our methods used only sentence or terminology knowledge. When sentence, terminology and template knowledge are used simultaneously, our method (*Term.+Sent.+Temp.*) outperforms the method (*Term.+Sent.*) using sentence and terminology knowledge by 2 BLEU on English-German and by 0.68 BLEU on average on English-Chinese on average respectively, which shows that the translation templates could effectively improve the translation performance.

| Metric | Method | Knowledge | English-German | | | | | English-Chinese | | | |
|---|---|---|---|---|---|---|---|---|---|---|---|
| | | | IT | Medical | Law | Koran | Avg. | Subtitles | News | Education | Avg. |
| BLEU | Priming-NMT | *Sent.* | 25.93 | 37.94 | 41.28 | 11.41 | 29.14 | 38.79 | 34.83 | 52.83 | 42.15 |
| | **Ours** | *Term.* | 25.50 | 31.88 | 35.24 | 10.33 | 25.74 | 25.32 | 33.67 | 41.47 | 33.49 |
| | | *Sent.* | 29.49 | 44.11 | 49.52 | 13.86 | 34.25 | 39.03 | 35.03 | 52.39 | 42.15 |
| | | *Term.+Sent.* | 31.34 | 43.74 | 49.77 | 13.61 | 34.62 | 39.95 | 37.53 | 53.73 | 43.74 |
| | | *Term.+Sent.+Temp.* | 32.43 | 45.14 | 50.00 | 18.89 | 36.62 | 40.13 | 37.87 | 55.25 | 44.42 |
| Exact Match | VecConstNMT | *Term.* | 80.91 | 83.99 | 83.87 | 77.27 | 81.51 | 92.31 | 88.01 | 94.39 | 91.57 |
| | **Ours** | *Term.* | 88.26 | 92.85 | 91.09 | 87.27 | 89.87 | 92.90 | 91.54 | 97.20 | 93.88 |
| | | *Sent.* | 39.75 | 56.83 | 57.72 | 23.64 | 44.49 | 52.94 | 57.20 | 61.11 | 57.08 |
| | | *Term.+Sent.* | 80.05 | 86.89 | 89.97 | 78.18 | 83.77 | 92.31 | 94.91 | 92.52 | 93.25 |
| | | *Term.+Sent.+Temp.* | 80.20 | 86.58 | 89.52 | 81.82 | 84.53 | 92.90 | 94.66 | 90.65 | 92.74 |

Table 4: Ablation result on BLEU and exact match accuracy using only partial type of knowledge on the English-German and English-Chinese.

| | $\lambda \geq 0.4$ | $\lambda \geq 0.5$ | $\lambda \geq 0.6$ |
|---|---|---|---|
| IT | 32.43 | 31.61 | 30.91 |
| Medical | 45.14 | 44.41 | 42.85 |
| Law | 50.00 | 49.10 | 47.76 |
| Koran | 18.89 | 17.56 | 16.38 |
| Avg | 36.62 | 35.67 | 34.46 |

Table 5: Effect of the threshold $\lambda$ for similar sentence retrieval on BLUE on the English to German.

| | IT | Medical | Law | Koran | Avg. |
|---|---|---|---|---|---|
| #Term. | 1.6 | 4.3 | 5.4 | 0.2 | 2.9 |
| #Sent. | 6.5 | 12.8 | 17.8 | 11.2 | 12.1 |
| #Temp. | 5.6 | 4.5 | 2.7 | 8.1 | 5.2 |
| #Know. | 13.7 | 21.6 | 25.9 | 19.5 | 20.2 |
| Speed | 1.6 | 1.9 | 1.9 | 1.7 | 1.8 |

Table 6: Relative inference speed for our method compared to the vanilla NMT in English to German multi-domain test sets. The batch size is 32. #Term., #Sent., #Temp., and #Know. indicate the average number of tokens of matching terminologies, similar sentences, predicted templates and whole knowledge sequences on the target, respectively.

## 4.4 Effect of Similarity Threshold $\lambda$

Most works (Bulte and Tezcan, 2019a; Xu et al., 2020; Pham et al., 2020) use 0.5 or 0.6 as a similarity threshold. Table 5 shows the effect of the threshold for similar sentence retrieval on translation quality. We find that retrieving similar sentences using a lower threshold leads to improvements. The average best performance on in-domain test sets can be achieved by our method based on the 0.4 threshold.

## 4.5 Inference Speed

We report the inference speed of our approach relative to the vanilla NMT in Table 6. We use the multiple types of knowledge as prefixes of the encoder and decoder of the NMT model, increasing the extra calculating time during decoding. The speed of the proposed method using beam search is 1.6~1.9 times slower than the vanilla NMT and mainly depends on the number of tokens of the prefixes on the target during decoding. Compared to $k$NN-MT with a generation speed that is two orders of magnitude slower than the vanilla NMT, our method is more easily acceptable in terms of inference speed.

## 5 Conclusions

In this paper, we propose a unified framework to integrate multi-knowledge into NMT models. We utilize multiple types of knowledge as prefixes of the encoder and decoder of NMT models, which guides the NMT model's translation process. Especially, our approaches do not actually require the model to see the domain-specific data in training. The model has automatic domain adaption capability and can be extended to new domains without updating parameters. The experimental results on multi-domain translation tasks demonstrated that incorporating multiple types of knowledge into NMT models leads to significant improvements in both translation quality and exact match accuracy.

Monolingual data is valuable to improve the translation quality of NMT models. In the future, we would like to integrate monolingual knowledge into the NMT model. Furthermore, our approach can be applied for tasks where there are multiple types of knowledge, such as Question Answering and Image to Text.

## Limitations

As with the majority of studies, the design of the current approach is subject to limitations. We integrate multiple types of knowledge as additional prefixes of NMT models and add time consumption in the training and inference stages. The experimental results show the added time cost of the proposed method is acceptable. Our approach depends on multiple types of knowledge and obtaining the knowledge may be difficult in some practical applications.

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
