# OpenReview forum: "Improving Neural Machine Translation by Multi-Knowledge Integration with Prompting"
_EMNLP/2023/Conference — EMNLP 2023 Findings_

### Official Review · Reviewer_wucw · 2023-08-04

**Soundness:** 3

**Excitement:**

2: Mediocre: This paper makes marginal contributions (vs non-contemporaneous work), so I would rather not see it in the conference.

**Paper Topic And Main Contributions:**

This paper is about improving neural machine translation (NMT) performance through multi-knowledge integration and prompting. The authors propose a unified framework that can effectively integrate multiple types of knowledge, including sentences, terminologies/phrases, and translation templates, into NMT models to guide the translation process. The approach requires no changes to the model architecture and can adapt to domain-specific translation without retraining. The experiments on English-Chinese and English-German translation demonstrate that the proposed approach outperforms baselines. The contributions of this work include proposing a simple and effective approach to integrate multi-knowledge into NMT models with prompting and demonstrating that an NMT model can benefit from multiple types of knowledge simultaneously.

**Reasons To Accept:**

1. This paper is relatively easy to understand.
2. The effect of this paper on English-German multi-domain translation is okay.
3. The inference speed of this method is acceptable.


**Reasons To Reject:**

1. The novelty of this paper is not enough and seems to be a supplement to some previous works (like Priming-NMT).
2. This paper does not show significant improvement over other methods in terms of the BLUE metric, and even has a noticeable gap compared to kNN-MT on the English-Chinese dataset.
3. Comparison to the latest methods in this paper is lacking. Additionally, the authors should demonstrate the performance of Language Model-based approaches such as LLaMA.


**Reproducibility:**

4: Could mostly reproduce the results, but there may be some variation because of sample variance or minor variations in their interpretation of the protocol or method.

**Reviewer Confidence:**

4: Quite sure. I tried to check the important points carefully. It's unlikely, though conceivable, that I missed something that should affect my ratings.

---

> ### Author Rebuttal · Authors · 2023-08-29
>
> Thank you for your feedback on our manuscript. Please let us address your concerns and provide answers to your questions.
> Our method is a supplement of previous work Priming-NMT.  The Priming-NMT retrieve similar source sentences and corresponding translations, and then the similar sentence pairs can be used as an input prompt for NMT models. First, we train the model based on our proposed loss function Equation2 and the NMT model is trained with standard loss function in their works. Also, our method can be applied to multi-knowledge and their work can only be is limited to sentences.  The performance of kNN-MT depends on training data quality, and Prompt-NMT achieves better BLEU on News and Education domains in table 3. Our method get better performance than fine-tuning when the domain data is noisy. Our approach has stronger generalization ability and significant performance improvements. We hope the above explanations could help to clarify part of your concerns.

---

### Official Review · Reviewer_GTCU · 2023-08-05

**Typos Grammar Style And Presentation Improvements:** n/a
**Soundness:** 2

**Excitement:**

2: Mediocre: This paper makes marginal contributions (vs non-contemporaneous work), so I would rather not see it in the conference.

**Missing References:**

n/a

**Paper Topic And Main Contributions:**

In this paper, the authors propose a unified framework for integrating multiple types of knowledge into Neural Machine Translation (NMT) models. By using the knowledge as prefixes for the encoder and decoder, the model is guided in learning how to use the knowledge effectively. This method is not domain-specific and does not require extra training for specific domains. Experiments on multi-domain translation tasks show significant improvements in translation quality and exact match accuracy.

**Questions For The Authors:**

n/a

**Reasons To Accept:**

Experimental results show significant improvements compared to baselines.
This paper is well written and the proposed method is easy to reproduced.

**Reasons To Reject:**

1. From the ablation experiments, the improvements brought by the proposed method mainly come from utilizing different types of external knowledge. However, other methods, such as kNN-MT, can also leverage this knowledge with simple modifications, making the comparison of experimental results insufficient.
2. The author needs to consider the performance of the proposed method when external knowledge contains noise.
3. As a training approach, the innovation of the proposed method is limited.
4. Large language models can easily incorporate different types of external knowledge into translation results. The author needs to prove the necessity of the method proposed in this paper.

**Reproducibility:**

5: Could easily reproduce the results.

**Reviewer Confidence:**

5: Positive that my evaluation is correct. I read the paper very carefully and I am very familiar with related work.

---

> ### Author Rebuttal · Authors · 2023-08-29
>
> Thank you for your feedback on our manuscript. Please let us address your concerns.
> We proposed method is to integate external knowledge into NMT models.  Compared with kNN-MT, our method achieving high translation quality in some domains. Our method get better performance than fine-tuning when the domain data is noisy. Our approach has stronger generalization ability and significant performance improvements. And we prove the importance of external knowledge to NMT models. We hope the above explanations could help to clarify part of your concerns.

---

### Official Review · Reviewer_m2EQ · 2023-08-06

**Soundness:** 3

**Excitement:**

3: Ambivalent: It has merits (e.g., it reports state-of-the-art results, the idea is nice), but there are key weaknesses (e.g., it describes incremental work), and it can significantly benefit from another round of revision. However, I won't object to accepting it if my co-reviewers champion it.

**Paper Topic And Main Contributions:**

This paper explores augmenting Neural Machine Translation by integrating multiple types of additional knowledge as prefixes in the model encoder and decoder, that the model can refer to to improve translation. The following types of knowledge are considered:

(1) similar source-target translation pair, if it exists, extracted from an in-domain parallel corpus, based on Edit Distance.

(2) translations of any specific terminology found in the source sentence, extracted from a bilingual dictionary, which itself is derived also from an in-domain parallel corpus.

(3) predicted translation template, which is generated by a model, trained to predict a pruned constituency tree in the target language, corresponding to the source sentence and source tree.

A baseline NMT model is trained on examples augmented with knowledge to learn to leverage it and then applied for translation inference. Although the use of each of those types of knowledge has been explored in previous papers, to the best of my knowledge, this is the first work that puts those methods in a common framework and reports results on combining all knowledge types in one model. Experiments are performed on En-De and En-Zh tasks and compared to relevant methods from the literature. BLEU scores and terminology accuracies are reported, showing improvement of the introduced method.

**Questions For The Authors:**

- Do you have results on En-Zh Spoken and Science domains? If not, consider dropping them from Table 1 for consistency.
- If I understand correctly, terminology match accuracy is based solely on matching terminology on the source and checking for its translation in the output, completely ignoring the reference translation. Are there cases where a terminology translation is missing in a reference translation?

**Reasons To Accept:**

- The method is straightforward and robust
- Moderately promising results

**Reasons To Reject:**

Although there is no major show stopper, there are a couple questionable points, that may be clarified based on authors response:

- Terminology match accuracy score seems to unfairly favour methods that include terminology knowledge, because the same terminology extraction method is both used during evaluation and to extract knowledge during model inference. Results in Table 3 show a huge difference between methods that use terminology constraints and those that don't. While I believe this still shows an actual improvement of terminology translation, ideally the structure of the evaluation method should not overlap with model architecture improvements to avoid biases to a particular terminology extraction method.
- Including translation template knowledge requires running an mBART model before the translation, which seems to not be clearly mentioned in the inference speed section. To avoid misleading conclusions, this point should be elaborated in the paper.

**Reproducibility:**

4: Could mostly reproduce the results, but there may be some variation because of sample variance or minor variations in their interpretation of the protocol or method.

**Reviewer Confidence:**

3: Pretty sure, but there's a chance I missed something. Although I have a good feel for this area in general, I did not carefully check the paper's details, e.g., the math, experimental design, or novelty.

**Typos Grammar Style And Presentation Improvements:**

- The point you are making in line 525 that this method may work better than fine-tuning when the data is noisy or there is a domain mismatch seems pretty important, it could be pointed our more explicitly.
- line 354: cloud -> could

---

> ### Author Rebuttal · Authors · 2023-08-29
>
> Thank you for your feedback on our manuscript. We apologize for the oversight regarding the typo you pointed out and will ensure its correction in the revised manuscript.
> First, The evaluation method of terminology is fair to the method VecConstNMT that include terminology knowledge. Our method and VecConstNMT use same terminology constraints in training and testing. We add Inference time of The mBART model in revised manuscript.   We remove  En-Zh Spoken and Science domains from Table 1 and thank you for reminding. The matching bilingual terminologies are from source sentences and its translations and the terminologies matched only to the source sentences  are ignored in our paper.
> We hope the above explanations could help to clarify part of your concerns. Thank you once again for your comments and suggestions to improve this paper.

---

### Meta-Review · Area_Chair_1Ho9 · 2023-09-26

**Recommendation:** 4

**Metareview:**

This paper introduces a method that mixes LLM-style prompting and seq2seq encoder-decoder methods in NMT. The result allows for combining terminology integration, few-shot translation and domain adaptation.

Pros: reviewers have relatively high opinions and raise valid but non-critical concerns.

Cons: the authors fail to address the raised concerns.

As a conclusion, the contribution is exciting, but there are major unaddressed questions with the paper: namely, that the terminology metric relies on the same terminology as the method (and thus additionally penalizes methods that do not use this terminology), no comparison to existing LLMs is given.

---

### Decision · Program_Chairs · 2023-10-07

**Decision:**

Accept-Findings

**Comment:**

This paper introduces a method that mixes LLM-style prompting and seq2seq encoder-decoder methods in NMT. The result allows for combining terminology integration, few-shot translation and domain adaptation.

Pros: reviewers have relatively high opinions and raise valid but non-critical concerns.

Cons: the authors fail to address the raised concerns.

As a conclusion, the contribution is exciting, but there are major unaddressed questions with the paper: namely, that the terminology metric relies on the same terminology as the method (and thus additionally penalizes methods that do not use this terminology), no comparison to existing LLMs is given.